# Sapropterin Dihydrochloride Responsiveness in Phenylketonuria: A Case Series Exploring Gaps in Comprehensive Patient Monitoring

**DOI:** 10.3390/nu17172892

**Published:** 2025-09-07

**Authors:** Manuela Lo Bianco, Roberta Leonardi, Alessia Migliore, Evelina Moliteo, Monica Sciacca, Sergio Rinella, Maria Grazia Pappalardo, Luisa La Spina, Marianna Messina, Riccardo Iacobacci, Martino Ruggieri, Concetta Meli, Agata Polizzi

**Affiliations:** 1Unit of Pediatric Clinic, A.O.U. Policlinico “G. Rodolico-San Marco”, P.O. “G. Rodolico”-University of Catania, 95123 Catania, Italy; lobianco.manuela@gmail.com (M.L.B.); mruggie@unict.it (M.R.); 2Postgraduate Residency Program in Pediatrics, Department of Clinical and Experimental Medicine, University of Catania, 95123 Catania, Italy; leonardi.roberta@outlook.it (R.L.); alessiamigliore16@gmail.com (A.M.); evelina.moliteo@gmail.com (E.M.); doc.monica.sciacca.ms@gmail.com (M.S.); 3Neonatal Intensive Care Unit (NICU), A.O.U. Policlinico “G. Rodolico-San Marco”, P.O. “G. Rodolico”-University of Catania, 95123 Catania, Italy; 4Unit of Pediatrics, Neonatology and Neonatal Intensive Care (NICU), San Vincenzo Hospital, 98039 Taormina, Italy; 5Department of Educational Sciences, University of Catania, 95131 Catania, Italy; sergio.rinella@hotmail.com; 6Unit of Expanded Neonatal Screening and Inherited Metabolic Diseases, Pediatric Clinic, Department of Medical Sciences, University of Catania, 95123 Catania, Italy; mariagrazia.pappalardo@policlinico.unict.it (M.G.P.); luisa.laspina@hotmail.it (L.L.S.); mmessina@unict.it (M.M.); riki713@hotmail.it (R.I.)

**Keywords:** phenylketonuria (PKU), sapropterin dihydrochloride, phenylalanine hydroxylase (PAH), BH4 loading test, international best practice

## Abstract

**Background**: Phenylketonuria (PKU) is a rare autosomal recessive metabolic disorder caused by mutations in the phenylalanine hydroxylase (*PAH*) gene, leading to hyperphenylalaninemia (HPA). Untreated, elevated phenylalanine (Phe) levels cause severe neurocognitive, developmental, and psychiatric complications. Management relies on a Phe-restricted diet, which is challenging to maintain, particularly in adolescents and adults. Sapropterin dihydrochloride, a synthetic form of tetrahydrobiopterin (BH4), can enhance residual PAH activity, lowering blood Phe levels and increasing dietary tolerance in responsive patients. However, real-world alignment with best practices remains underexplored. This study aims to report a tertiary referral center’s experience with sapropterin treatment in PKU and assess adherence to international guidelines. **Methods**: We retrospectively analyzed 23 PKU patients treated with sapropterin from 2007 to 2025. Patients with baseline Phe levels of 360–2000 µmol/L underwent a 10 mg/kg/day loading test over two weeks. Responsiveness was defined as a ≥30% reduction in blood Phe levels. Phe levels were measured pre- and post-test, and dietary tolerance was evaluated. Adherence to best practices was critically reviewed. **Results**: All patients showed significant Phe reductions (mean 71.43%, *p* < 0.0001), exceeding responsiveness thresholds. Most achieved substantial increases in dietary Phe tolerance, with three patients partially responsive (800–1200 mg/day). Responsiveness was unrespectful of the patient’s genotype, for those individuals for whom this was known (8/23 patients). Although effective, the test dose and duration differed from guideline recommendations (20 mg/kg/day). Neuropsychological and QoL assessments were not systematically performed, representing a key limitation. **Conclusions**: Sapropterin dihydrochloride effectively identified responders and improved dietary flexibility even with lower dosing protocols. Greater adherence to international standards, particularly regarding long-term neuropsychological monitoring, is needed to optimize patient care.

## 1. Introduction

Phenylketonuria (PKU, MIM #261600) is an autosomal recessive inborn error of metabolism with a prevalence varying from 1 in 10,000 births in Europe to higher rates in Turkey and Ireland and lower rates in Finland [1,2,3]. The disease is caused by mutations in the *PAH* gene, which encodes the enzyme phenylalanine hydroxylase. This enzyme is responsible for converting the amino acid phenylalanine (Phe) to tyrosine (Tyr) in the presence of cofactors such as tetrahydrobiopterin (BH4), oxygen, and iron. Reduced or absent enzyme activity due to mutations in the *Phenylalanine hydroxylase* gene (PAH, #612349) leads to accumulation of phenylalanine in blood and tissues, which is toxic to the developing brain. Over 1000 PAH mutations are known with varying degrees of severity. About 60% of them are missense mutations reducing or abolishing enzyme activity through protein misfolding or impaired catalysis [1,2,3,4,5,6].

If untreated, PKU can result in severe intellectual disability, seizures, behavioral problems, and microcephaly. Early diagnosis through newborn screening and immediate dietary management with a phenylalanine-restricted diet can prevent neurological damage. Tyrosine becomes an essential amino acid in PKU patients and must be supplemented. PKU is one of the most well-known treatable metabolic disorders. Advances in therapy include Sapropterin (the BH4 cofactor) and gene therapy research. Management requires lifelong adherence and monitoring to optimize neurocognitive outcomes [3,4,5,6].

The classification of PKU has evolved, moving from simplistic to more nuanced systems that combine biochemical, genetic/molecular, and clinical data [7,8].

Modern PKU is classified by blood Phe levels at diagnosis and by Phe tolerance-defined as the daily Phe intake that maintains blood levels within therapeutic targets: 120–360 μmol/L in children under 12 and pregnant women, and 120–600 μmol/L in individuals over 12 years. Based on Phe tolerance, four phenotypes are distinguished: classical PKU (<20 mg/kg/day), moderate (20–25 mg/kg/day), mild (25–50 mg/kg/day), and mild HPA (no diet required) [9,10]. Tolerance is influenced by genotype, adherence to diet, protein intake, and responsiveness to pharmacological treatments with BH4 analogues such as sapropterin dihydrochloride [11,12,13].

According to the recently published first revision of the European Guidelines on PKU classification should not rely solely on untreated Phe levels but also integrate responsiveness to cofactor therapy, reflecting a more nuanced and clinically relevant approach to patient management. Thus, patients should be classified as: (a) not requiring treatment (Phe < 360 μmol/L); (b) requiring treatment, cofactor responsive; or (c) requiring treatment, cofactor unresponsive [14].

BH4 responsiveness refers to the ability of a PKU patient to respond to treatment with tetrahydrobiopterin (BH4), which is a natural cofactor of the enzyme phenylalanine hydroxylase (PAH). In some PKU patients, especially those with residual PAH activity, administering pharmacological doses of BH4 (like sapropterin dihydrochloride) can enhance the activity of the faulty enzyme. This leads to lower blood phenylalanine (Phe) levels, improved tolerance to dietary Phe, and sometimes even the possibility of a more relaxed diet [11,15,16].

However, not all PKU patients respond, so a BH4 loading test is now performed to identify responders vs. non-responders. Responsiveness often depends on the type of *PAH* mutation a patient has [17]. Thus, genetic/molecular classification based on the specific mutations in the *PAH* gene could help predict enzyme residual activity and potential BH4 responsiveness. Moreover, the emerging functional classification considers metabolic tolerance, neurocognitive risk, and treatment need over time, aiming for a personalized medicine approach rather than rigid biochemical thresholds [15,18].

Currently, the lack of uniform classification contributes to variability in diagnosis, treatment thresholds, and use of BH4, and long-term management strategies are often worsened by resource disparities and limited access to proper multidisciplinary care. This is why many international experts are calling for harmonized, evidence-based guidelines and/or best practices for PKU [19].

By reporting the present personal case-series of individuals with PKU, the authors would like to further address current gaps and discrepancies in PKU classification and management. Following the characterization of the clinical, biochemical, and treatment profiles of patients with PKU who demonstrated responsiveness to Sapropterin dihydrochloride and subsequently underwent dietary liberalization, the investigators evaluated adherence to current international guidelines for the assessment and management of BH4 responsiveness and identified areas of variability or deviation in local practice. Insights that may inform efforts toward greater standardization and harmonization of care in PKU are provided.

## 2. Materials and Methods

This retrospective study reviewed the medical records of 38 individuals diagnosed at birth as having mild to moderate phenylketonuria (PKU) and followed at the Center for Metabolic and Rare Diseases of the Pediatric Clinic, University of Catania, Catania, Italy, between December 2007 and January 2025. Soon after the diagnosis, all patients underwent a Sapropterin dihydrochloride (BH4) loading test; only those who exhibited a positive response were included in the final analysis.

Patients with classical PKU carrying biallelic null mutations in the *PAH* gene, which are known to confer Sapropterin non-responsiveness, were excluded from the study.

Enrolled patients had baseline blood phenylalanine (Phe) levels ≥ than 360 μmol/L, in accordance with established guidelines. Blood Phe levels were assessed before the test, daily during the test, and for two weeks after. Body weight was also recorded for dose determination, and dietary Phe tolerance was assessed both before and after treatment.

Phenylalanine levels were assayed through dried blood spot (DBS) samples and measured using flow-injection electrospray-tandem mass spectrometry (ESI-MS-MS). These measurements were conducted in multiple reaction monitoring mode (MRM), employing stable isotope-labeled internal standards for quantitative analysis. These samples were daily collected under the same conditions, with patients fasting, at the same time of day and performed in an outpatient setting to minimize inconvenience to patients and families.

Before sapropterin initiation, all patients were managed with an individually tailored Phe-restricted diet, in line with their metabolic tolerance. Dietary prescriptions included natural foods with low Phe content (mainly fruits and vegetables), with or without limited amounts of natural protein sources, depending on individual tolerance. To ensure adequate protein and micronutrient intake, patients received Phe-free amino acid mixtures as protein substitutes, as well as multivitamin and long-chain polyunsaturated fatty acid (LC-PUFA) supplementation.

In the week preceding the Sapropterin loading test, dietary intake was gradually adjusted to determine each patient’s maximum individual phenylalanine (Phe) tolerance, while ensuring that pre-test blood Phe levels remained within the range of 1200–2000 μmol/L.

This meant approximately doubling their daily phenylalanine intake (e.g., from 800 mg/day to 1600–1800 mg/day), achieved through a free diet when possible or by incorporating foods such as legumes, dairy products, and occasionally ham or sausages. On the eighth day, the Sapropterin loading test began at a dose of 10 mg/kg/day for two weeks, during which the same diet was maintained. Responsiveness to the Sapropterin loading test was defined, in accordance with international guidelines, as a reduction of ≥30% in blood phenylalanine levels from baseline during Sapropterin administration [19].

After initiation of Sapropterin, dietary management was adjusted according to the response to treatment. A diet was considered “free” for PKU patients when their dietary Phe tolerance exceeded 2000 mg/day, allowing them to maintain blood Phe levels within therapeutic targets (120–360 μmol/L) without significant restrictions on Phe intake.

For responders, therapy with Sapropterin was initiated at a dosage between 5 and 20 mg/kg/day, depending on blood Phe levels and individual tolerance.

The occurrence of any clinical events associated with Sapropterin dihydrochloride during both the administration of the loading test and the subsequent treatment phase has been recorded in order to monitor the safety and tolerability of the treatment. Evaluation included gastrointestinal disturbances (abdominal pain, diarrhea, and vomiting), upper respiratory tract infections, nasal congestion, and headaches.

To ensure accurate test execution, families received detailed instructions on the test protocol, emphasizing the importance of consistent blood sample collection and strict adherence to the prescribed treatment regimen. They were also informed that while treatment with Sapropterin may not always enable a completely unrestricted diet for PKU patients, it can substantially enhance dietary Phe tolerance, leading to greater dietary flexibility and an improved quality of life.

Moreover, all patients (or their parents, in the case of children) were accustomed to completing a daily food diary, which was systematically brought to each follow-up visit and reviewed together with the clinical team. These diaries included information on food choices, dietary preferences, and any issues related to the acceptance or palatability of Phe-free protein substitutes. This approach allowed clinicians to adapt the dietary regimen to the individual needs and tastes of each patient, ensuring both adherence and personalization of nutritional management.

Finally, molecular data, including details on zygosity, allelic variants, corresponding amino acid substitutions, and references to the Blau database or relevant scientific literature, were reported.

### Statistical Analysis

Stata/SE 18.0 software was used to perform statistical analysis. To assess the effectiveness of sapropterin dihydrochloride treatment in reducing Phe levels in our series of patients, we calculated the percentage difference in Phe levels before and after two weeks of treatment with Sapropterin using the following formula:*Percentage Difference* = *Phenylalaninemia pre-sapropterin* − *Phenylalaninemia post-sapropterin*/*Phenylalaninemia pre-sapropterin*) × 100.

Before proceeding with inferential analyses, we examined the data distribution for the variables “*Phenylalaninemia pre-sapropterin*”, “*Phenylalaninemia post-sapropterin*”, and “*Percentage Difference*” to check for normality using the Shapiro–Wilk test. Having observed a non-normal distribution in the “*Phenylalaninemia pre-sapropterin*” and “*Phenylalaninemia post-sapropterin*” variables, we applied the Wilcoxon signed-rank test for paired samples to compare pre- and post-treatment Phe levels. For the “*Percentage Difference*” variable, which showed a normal distribution, we conducted a one-sample *t*-test to test if the mean percentage reduction was significantly greater than 30%. Alpha was 0.05.

The testing protocol implemented at our centre was compared with the 2018 international best practice guidelines for assessing responsiveness to Sapropterin dihydrochloride [18]. Adherence to these recommendations was critically evaluated, and any deviations were analysed with particular attention to their potential underlying causes, practical limitations, and implementation challenges.

## 3. Results

### 3.1. Patients’ Characteristics, Sapropterin Dihydrochloride Responsiveness, and Dietary Tolerance

Among the 38 patients tested with Sapropterin dihydrochloride between 2007 and 2025, 23 were found to be responsive to Sapropterin at a dose of 10 mg/kg/day and were included in this study. The remaining 15 patients were classified as non-responders. Among them, 3 underwent a further loading test at an increased dose of 20 mg/kg/day, as requested by their parents; however, none demonstrated a positive response.

Responders comprised 10 males and 13 females, with a mean age of 13.39 years (SD ± 8.21). The mean Phe level at birth, as measured through newborn screening, was 602.73 µmol/L (SD ± 201.99 µmol/L). The median pre-test Phe level was 484.29 µmol/L (IQR 484.29–726.44), while at two weeks post-test it was 157.39 μmol/L (IQR 121.07–181.61).

Although the Sapropterin loading test was administered over a period of two weeks, in all responsive cases, a significant reduction in blood Phe levels (≥30% from baseline) was observed within 48–72 h of initiating treatment. This early response was consistent across all responders. Data analysis revealed that the average percentage reduction in Phe levels after two weeks of Sapropterin administration at the dosage of 10 mg/kg/day was 71.43% (95% CI: 68.61–74.25). The Shapiro–Wilk tests indicated a non-normal distribution for pre-treatment Phe levels (W = 0.856, *p* = 0.0035) and post-treatment levels (W = 0.814, *p* = 0.0007), but an approximately normal distribution for the percentage difference (W = 0.924, *p* = 0.082). The Wilcoxon signed-rank test for paired samples confirmed a significant reduction in Phe levels (*p* < 0.05) after the loading test. The one-sample *t*-test for the “Percentage Difference” variable showed a significant reduction in Phe levels exceeding the established 30% threshold (t = 30.447, *df* = 22, *p* < 0.0001), confirming the treatment’s efficacy (Table 1).

None experienced collateral events or intercurrent sickness during the test that necessitated discontinuation.

Most responders were treated at a dosage of 10 mg/kg/day, except for 2 patients who received 5 mg/kg/day, as indicated in Table 2.

Following treatment, 21 out of 23 patients achieved sufficient phenylalanine (Phe) tolerance to discontinue dietary restrictions. The remaining two patients exhibited partial responsiveness: patients 7 and 13 achieved tolerances of 1200 mg/day (Table 2) and continued on a partially Phe-restricted diet, consisting of about one-third Phe-free protein substitutes and two-thirds natural protein sources. In these two cases, the Sapropterin dosage was not increased beyond 10 mg/kg/day, as both were enrolled in the international KAMPER trial (BioMarin, NCT01016392), which restricted the maximum permitted dose to 10 mg/kg/day.

### 3.2. Molecular Data and Sapropterin Responsiveness

Table 3 shows details on zygosity, allelic variants, and amino acid substitutions together with references to Blau’s database or the scientific literature.

Molecular data were obtained in 9/23 individuals; among these, 7 unique mutations were identified (Table 3). In one case, the identified mutation had not been previously evaluated for Sapropterin responsiveness in the scientific literature. The genotypes c.898G>T (p.Ala300Ser) and c.1223G>A (p.Arg408Gln), found in patients 17 and 18, respectively, have been previously reported in three cases in the literature. However, Sapropterin responsiveness had not been evaluated in those earlier reports. In the present study, both were tested and found to be responders.

Patient 2 carried the genotype c.826A>G (p.Met276Val) and c.1066-11G>A (IVS10-11G>A), which had not been previously reported in the literature nor assessed for Sapropterin responsiveness. In our study, this patient demonstrated a positive response to treatment.

For the remaining patients with available molecular data, the identified genotypes were more commonly reported in the literature and had previously been tested for Sapropterin responsiveness, consistently demonstrating a positive response. Molecular information was unavailable for the rest of the series.

### 3.3. Adherence to 2018 Best Practice Guidelines for Sapropterin Testing and Treatment

Our clinical practice was critically compared with the 2018 International guidelines for evaluating responsiveness to Sapropterin dihydrochloride [19], revealing overall alignment with only minor deviations. Notably, in our series, the loading test was performed using a dosage of 10 mg/kg/day, instead of the 20 mg/kg/day recommended by the guidelines.

Despite the lower dosage, the 10 mg/kg/day regimen was sufficient to demonstrate responsiveness in all responders within our series. In three non-responsive patients, Sapropterin was subsequently administered at the higher dose of 20 mg/kg/day—per parental request—but no clinical benefit was observed. Additionally, the duration of our loading test was two weeks, which falls within the guideline-recommended range of 48 h to four weeks, confirming adherence to established testing protocols.

Baseline data, including pre-treatment blood Phe levels, dietary Phe tolerance, and body weight for dose determination, were systematically gathered following established guidelines. Additionally, parents were provided with thorough education regarding the test requirements and expectations to ensure accurate and reliable implementation.

Finally, our patients underwent neuropsychological evaluations during follow-up assessments. However, standardized neuropsychological, behavioural, or quality-of-life evaluations were not consistently or serially administered, either before or after treatment with Sapropterin, which contrasts with the recommendations outlined in international guidelines [19].

## 4. Discussion

Phenylketonuria (PKU) is one of the most common inborn errors of metabolism [20]. Affected children typically show no evident clinical abnormalities at birth. Physical characteristics that may be present are blond hair, blue eyes, reduced growth, microcephaly and a distinctive mousey odor. If untreated, these children can develop epilepsy, tremors, limb spasticity, and intellectual disabilities within the first year of life [21]. Since the introduction of neonatal screening for PKU in the early 60s [22], clinical manifestations have become less frequent as timely therapeutic interventions have consequently been implemented.

The primary goal of PKU therapy is to control the concentration of Phe in the blood and, consequently, in the brain, thus preventing intellectual disability, behavioral problems and ensuring normal neurocognitive and psychosocial development. The cornerstone of treatment involves a phenylalanine-restricted diet [1]. However, adhering to this diet lifelong proves extremely challenging. While dietary adherence is meticulous in childhood, it tends to wane during adolescence and adulthood, likely due to increased patient independence and the psychological and social burdens of maintaining a strict diet. Despite advancements in the taste and smell of dietary supplements, many individuals with PKU continue to experience a reduced quality of life. This is because the daily management of the disorder often still requires strict dietary restrictions, frequent monitoring, and social limitations, which can be physically and emotionally challenging—even if the supplements themselves have become more palatable [23].

In recent years, therapeutic perspectives in PKU and Hyperphenylalaninemia (HPA) have expanded, and pharmacological treatments have been developed, offering these patients possibilities for a better quality of life (QoL). Since 1999, it has been widely reported that pharmacological doses of tetrahydrobiopterin (BH4), the cofactor for phenylalanine hydroxylase (PAH) enzyme, can reduce blood Phe levels in many PKU patients, allowing for dietary relaxation or even cessation [9,21].

Sapropterin dihydrochloride, a pharmaceutical formulation of BH4, was approved in 2007 by the Food and Drug Administration (FDA) for use in the United States and by the European Medicines Agency (EMA) in 2008 for the treatment of a subset of PKU patients across all ages. In Italy, Sapropterin dihydrochloride received approval from the Italian Medicines Agency (AIFA) in 2017, further expanding its availability for clinical use. According to the 2018 International best practice for the evaluation of responsiveness to Sapropterin dihydrochloride in patients with PKU, responsiveness is defined as a reduction of ≥30% in blood Phe levels from baseline. Reduction should be observed after administering a single 20 mg/kg/day dose of sapropterin dihydrochloride over a minimum test duration of 48 h. Longer trials of up to four weeks may be necessary to assess changes in dietary Phe tolerance or to confirm responsiveness in certain cases. In general, a prolonged test is preferable, and a 48 h test followed by a 1-week extension should only be performed when a longer initial test is not feasible. The extension is applied if there is no immediate reduction of over 30% in blood Phe levels [10].

It is worth noting that responsiveness to Sapropterin is influenced by various factors, including the *PAH* genotype, which impacts the residual activity of the PAH enzyme. According to current European guidelines, the genotype is highly predictive of BH4 responsiveness, which, therefore, should be evaluated in all patients before starting treatment. This allows for the prediction of response in advance, especially if the response to Sapropterin for the specific genotype has already been reported [1,24,25,26].

Sapropterin dihydrochloride response testing is recommended for patients with untreated baseline blood Phe levels between 360 and 2000 μmol/L. Blood Phe targets for PKU management differ between European and American guidelines. European recommendations suggest maintaining Phe levels below 360 μmol/L until age 12, while American guidelines advocate for lifelong levels between 120 and 360 μmol/L [1,27]. Sapropterin dihydrochloride can help reduce the burden of a phenylalanine-restricted diet, even in patients who are already maintaining target blood Phe levels. Response testing should be offered to individuals who were not tested at birth, and previous neonatal testing should not preclude re-evaluation—particularly in cases with genotypes associated with potential BH4 responsiveness—as initial testing may fail to identify slow responders [18,28,29,30]. Ideally, diagnostic Phe concentrations should guide the decision to initiate testing, as they more accurately reflect the patient’s current metabolic status compared to neonatal screening values. However, in cases where diagnostic testing is delayed, it is acceptable to begin Sapropterin treatment based on screening Phe levels to prevent unnecessary treatment delays. If subsequent diagnostic Phe levels are found to be below 360 μmol/L, Sapropterin can be discontinued. Moreover, diagnostic evaluations may uncover alternative conditions—such as BH4 metabolism disorders, liver diseases, or DNAJC12 deficiency—that require different therapeutic strategies [31].

According to current guidelines, if the loading test indicates an increased tolerance to Phe, Sapropterin dihydrochloride therapy should be continued at the lowest effective dose, typically within the range of 5 to 20 mg/kg/day [19].

In this study, we evaluated 23 patients with phenylketonuria (PKU) who underwent a pre-treatment loading test with Sapropterin dihydrochloride, demonstrated a positive response, and subsequently received ongoing treatment. We analyzed their blood phenylalanine (Phe) levels before and after treatment, along with their dietary Phe tolerance. When available, we also included genotype information and noted whether the associated Sapropterin responsiveness had been previously reported. Interestingly, only one patient carried a genotype for which responsiveness to Sapropterin had not been previously documented. Additionally, we critically reviewed our clinical approach in relation to current international best practice guidelines for Sapropterin testing and treatment [19].

All patients in our case series showed a positive response to the loading test, which was conducted over two weeks at a dose of 10 mg/kg/day. Our findings indicate that effective responsiveness can be identified without the need to escalate to the guideline-recommended dose of 20 mg/kg/day or extend the test duration to four weeks. Similarly, Matalon et al. [30] reported that a single 10 mg/kg dose was sufficient to identify the majority of BH4-responsive patients, with 58% achieving a significant (>30%) reduction in blood Phe within 24 h. Vernon et al. further supported these results, describing a two-step dosing approach in which an initial dose of 10 mg/kg/day successfully identified most responders within 7 days. Notably, 62% of patients were classified as responders in that study, although some non-responders required an increased dose of 20 mg/kg/day for further evaluation [32,33].

These findings support the notion that lower Sapropterin doses may effectively identify responders, and that a two-week testing period could represent a practical balance between the minimum 48 h and maximum four-week durations recommended. A shorter test duration offers several advantages: it enables earlier detection of responsiveness, improves patient adherence, and lowers treatment costs. Additionally, it may reduce the psychological stress experienced by patients and families, minimize the risk of dropouts, and allow for more efficient use of clinical resources—especially in pediatric populations. This streamlined approach also increases overall testing capacity, making it both cost-effective and operationally sustainable. Extending the duration beyond two weeks should be considered only in cases where treatment response is ambiguous or delayed.

In accordance with international guidelines [19], all patients in our study initiated Sapropterin treatment at doses ranging from 5 to 20 mg/kg/day—most commonly at 10 mg/kg/day, with two patients starting at 5 mg/kg/day. This dosing strategy allowed the majority of patients to achieve sufficient dietary Phe tolerance to transition to an unrestricted diet. However, two individuals continued to require dietary restrictions due to only partial responsiveness. Importantly, no significant adverse events were reported during either the loading phase or the subsequent treatment period, further supporting the well-established safety profile of Sapropterin dihydrochloride [34,35,36].

From a nutritional perspective, our findings underscore the substantial impact of sapropterin therapy on food intake and dietary management in PKU. Before treatment, all patients required a strict Phe-restricted diet, supported by Phe-free amino acid mixtures and micronutrient supplementation. Following responsiveness to Sapropterin, most patients were able to liberalize their diet, discontinue protein substitutes, and rely predominantly on natural food sources, thus improving dietary quality of life.

To gain a comprehensive understanding of the long-term clinical benefits of Sapropterin dihydrochloride, ongoing monitoring should extend beyond simple blood Phe concentrations. It should also account for fluctuations over time and the Phe-to-tyrosine (Tyr) ratio, in conjunction with assessments of dietary Phe tolerance. Evaluating long-term effects requires the gradual and guided increase in dietary Phe intake, following established clinical recommendations. Additionally, it is essential to track any changes in the use or dependency on specialized low-protein foods and amino acid-based nutritional products, including their tolerability and any associated challenges.

For this reason, our patients undergo regular monitoring of blood phenylalanine (Phe) levels, typically on a weekly to monthly basis, along with assessments of the Phe-to-tyrosine (Tyr) ratio. These parameters are used to evaluate and adjust their individual dietary Phe tolerance. In addition, patients receive clinical follow-up visits every 6 to 12 months to ensure the ongoing effectiveness and safety of their treatment regimen and to address any evolving clinical needs.

Considering the significant impact of the disease on patients’ quality of life, as well as its effects on neurocognitive and behavioural aspects, it is essential to follow best practices through regular follow-up. This should include routine neurocognitive testing and periodic reassessments of quality of life, particularly in relation to the therapeutic options available. At our center, we have evaluated the psychological and behavioral profiles of these patients, as well as their quality of life, both before and after administering the drug. However, these assessments were not carried out systematically, nor did they use standardized testing tools, which has limited the objectivity and consistency of the results.

This underscores the need for a more structured and organized approach, as highlighted in the study by Feldmann et al. [37]. In their research, the use of standardized neuropsychological tools, such as the Wechsler Intelligence Scales for Children (WISC-IV) and the Wechsler Adult Intelligence Scale (WAIS-IV) [38], enabled a more objective and measurable evaluation of cognitive outcomes. Their findings stress that, in addition to traditional assessments, monitoring variations in blood phenylalanine (Phe) levels and their relationship with cognitive development—especially in patients treated with Sapropterin dihydrochloride—provides valuable insights. The study revealed significant improvements in processing speed and full-scale IQ in patients treated with Sapropterin, compared to those on a classic Phe-restricted diet, whose full-scale IQ remained stable with only minor improvements in cognitive subdomains.

Similarly, the findings from Grant et al. [39] highlight the significance of systematic monitoring, demonstrating that treatment with Sapropterin dihydrochloride led to substantial improvements in attention and executive functioning in children and adolescents with PKU, as assessed by the ADHD RS-IV and BRIEF scales [26]. Both studies emphasize the crucial role of metabolic control in enhancing cognitive and behavioral outcomes, further supporting the idea that lowering Phe levels is essential for alleviating neuropsychiatric impairments in PKU. These findings underscore the potential cognitive benefits of Sapropterin dihydrochloride treatment, likely due to its ability to improve dietary flexibility and metabolic control. They also reinforce the necessity for standardized and systematic neuropsychological evaluations to objectively assess treatment outcomes and gain a deeper understanding of their effects on this patient population.

## 5. Limitations

The primary limitations of our study include the relatively small sample size of 23 patients, which may restrict the generalizability of our findings. Additionally, molecular data were not available for all patients, confining our ability to analyze the relationship between specific genetic variants and responsiveness to Sapropterin treatment. Furthermore, our study did not incorporate systematic neuropsychological, behavioral, or quality-of-life (QoL) assessments before or after treatment, as recommended by International Guidelines. These assessments are essential for gaining a comprehensive understanding of the broader clinical benefits of treatment with Sapropterin, particularly its effects on cognitive and psychosocial outcomes. Finally, while our study demonstrated the efficacy of a two-week loading test with doses of 10 mg/kg/day, this lower starting dose, in comparison to the guideline-recommended 20 mg/kg/day, represents a deviation that requires further investigation.

## 6. Conclusions

Despite international guidelines recommending a loading test dose of 20 mg/kg/day, the patients in our study demonstrated responsiveness at lower doses of 10 mg/kg/day. Our findings confirm that Sapropterin dihydrochloride is effective in significantly reducing blood Phe levels and increasing dietary tolerance in PKU patients. According to our experience, administering Sapropterin at these lower doses during a two-week loading test was sufficient to identify responders. This also facilitated dietary liberalization and improved metabolic control. Comprehensive neuropsychological assessments with dedicated tests are needed to explore the long-term effects of Sapropterin on cognitive outcomes, neuropsychiatric improvements, and quality of life. Nonetheless, this approach is still far from being adopted regularly in clinical practice. The absence of standardized neuropsychological assessments in local PKU care practices can be attributed to several interrelated factors. Many healthcare settings, particularly in resource-limited regions, face challenges such as a shortage of specialized professionals. This scarcity hampers the implementation of comprehensive neuropsychological evaluations for PKU patients. In some countries, while national protocols for PKU management exist, they may not provide specific recommendations on the appropriate neuropsychological assessment tools to use. This gap leads to inconsistent practices and reliance on non-standardized methods, which can compromise the objectivity and reliability of evaluations. Moreover, social determinants like low health literacy can deter patients from seeking comprehensive care, especially in long-term monitoring. These barriers contribute to missed opportunities for thorough neuropsychological monitoring and follow-up. The underestimation of neuropsychological issues in PKU patients can lead to the neglect of routine cognitive and behavioural assessments, despite evidence linking elevated blood phenylalanine levels to cognitive deficits. Overall, while some deviations from international best practices were noted, our protocol aligns closely with established recommendations and demonstrates the feasibility of adapting guidelines to local resource availability and experience. However, overcoming the current challenges requires a comprehensive approach, which includes the allocation of resources and increasing awareness about the importance of neuropsychological monitoring in the management of PKU.

## Figures and Tables

**Table 1 nutrients-17-02892-t001:** Phenylalaninemia Levels Pre- and Post-Sapropterin Treatment and Phenylalaninemia Percentage Reduction. This table presents data from our series of 23 patients, detailing Phe levels measured before and after two weeks of treatment with sapropterin dihydrochloride, as well as the Phe percentage reduction observed in each case. The median and Interquartile Range (IQR) of these reductions are provided at the bottom of the table, as well as the mean and standard deviation (SD). Statistical analysis was conducted using the Wilcoxon test to compare pre- and post-treatment Phe levels, yielding a significant *p*-value (<0.05). Additionally, a one-sample *t*-test was applied to the percentage reductions, demonstrating a significant decrease in Phe levels, which exceeded the 30% threshold (*p*-value < 0.0001).

Patients ID (Sex)	Phenylalaninemia Levels Pre-Sapropterin Loading Test (µmol/L)	Phenylalaninemia Levels Post-Sapropterin Loading Test (µmol/L)	Phenylalaninemia Levels Reduction Post-Sapropterin Loading Test (%)
1 (M)	1089.65	242.22	77.67
2 (F)	908.05	181.61	80.00
3 (F)	726.44	181.61	75.00
4 (M)	1089.70	242.15	77.78
5 (F)	484.29	121.07	75.00
6 (M)	484.29	151.34	68.75
7 (F)	726.44	181.61	75.00
8 (M)	484.29	121.07	75.00
9 (M)	363.22	121.07	66.67
10 (F)	484.29	121.07	75.00
11 (F)	502.45	181.61	63.85
12 (F)	484.29	121.07	75.00
13 (F)	726.44	272.41	62.50
14 (F)	605.36	199.77	67.00
15 (F)	484.29	181.61	62.50
16 (F)	484.29	181.61	62.50
17 (F)	484.29	169.50	65.00
18 (M)	484.29	157.39	67.50
19 (M)	1089.70	157.39	85.56
20 (M)	484.29	121.07	75.00
21 (M)	484.29	121.07	75.00
22 (M)	363.22	121.07	66.67
23 (F)	605.36	121.07	80.00
Median (IQR) Mean (±SD)	484.29 (484.29–726.44) -	157.39 (121.07–181.61) -	- 71.43 (±6.52)
*p*-value	(Wilcoxon signed-rank test) <0.05		(One-sample *t*-test) <0.0001

**Table 2 nutrients-17-02892-t002:** Dietary Tolerance: This table summarizes the dietary tolerance data for 23 PKU patients treated with sapropterin at our Center for Metabolic and Rare Diseases. It includes the pre-treatment dietary tolerance (mg/day) and the post-treatment dietary tolerance, indicating whether the patients achieved free dietary tolerance (defined as tolerance exceeding 2000 mg/day) or showed improvements in their tolerance levels. Additionally, the table reports the dosage of sapropterin administered during treatment following the test.

Patient	Pre-Treatment Phe Tolerance (mg/day)	Post-Treatment Phe Tolerance (mg/day)	Sapropterin Dihydrochloride Dosage (mg/kg/day)
1	600	Free	10
2	800	Free	10
3	800–1000	Free	10
4	600	Free	10
5	1000	Free	10
6	1000	Free	10
7	600	1200	10
8	1000	Free	5
9	600	Free	10
10	600	Free	10
11	600–700	Free	10
12	600–700	Free	10
13	600	1200	10
14	600–800	Free	10
15	700	Free	10
16	800	Free	10
17	800	Free	10
18	800	Free	10
19	600–700	Free	10
20	800	Free	10
21	900	Free	5
22	1000	Free	10
23	800	Free	10

**Table 3 nutrients-17-02892-t003:** The table summarizes the six genotypes identified among patients in our case series. For each individual, it presents the specific allelic variants and corresponding amino acid substitutions, the zygosity, and references to whether the genotype has been previously reported in Blau’s database or the broader scientific literature. It also indicates whether the genotype had been previously tested for Sapropterin responsiveness and the corresponding outcomes. An asterisk (*) next to patient 2 denotes that both the genotype and its associated Sapropterin responsiveness are being reported for the first time in this study. Patients 17 and 18, whose genotypes had been previously described in three cases but had not been assessed for responsiveness, were tested in our cohort and demonstrated a positive response to Sapropterin.

Patients ID	Allelic Variant 1 (Amino Acid Substitution)	Allelic Variant 2 (Amino Acid Substitution)	Zygosity (HTZ/ HMZ)	No. Reported Patients (Blau’s Database/ Literature)	Tested for Sapropterin Response/Sapropterin Testing Status	Responsive to Sapropterin
1	c.1028A>G (Y343C)	c.829T>G (Y277D)	HTZ	2	2	1
2	c.826A>G (M276V)	c.1066-11G>A (IVS10-11G>A)	HTZ	0	1 *	1 *
3	c.143T>C (L148S)	c.1241A>G (Y414C)	HTZ	5	3	3
4	c.1028A>G (Y343C)	c.829T>G (Y277D)	HTZ	2	2	1
5	c.143T>C (L148S)	c.143T>C (L148S)	HMZ	94	58	43
6	c.631C>A (P211T)	c.782G>A (R2Q61)	HTZ	11	8	8
17	c.898G>T (A300S)	c.1223G>A (R408Q)	HTZ	3	1 *	1 *
18	c.898G>T (A300S)	c.1223G>A (R408Q)	HTZ	3	1 *	1 *

## Data Availability

The data supporting the reported results are available at the University Hospital “Policlinico-San Marco”, Catania, Italy.

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
