# Peer review of "Sapropterin Dihydrochloride Responsiveness in Phenylketonuria: A Case Series Exploring Gaps in Comprehensive Patient Monitoring"

_nutrients, 2025, doi:10.3390/nu17172892_

Round 1
Reviewer 1 Report
Comments and Suggestions for Authors
the manuscript deals with an important topic, it is of interest for PKU scientific community.
Major question:
Table 1:
How do you explain that the pre- and post-therapy blood Phe levels showed similar values in most patients (e.g. 484.29 or 726.44 micromol/L)? In reality, there are no such similar blood values in a PKU group. This is a significant question, because the main achievements were concluded from these blood results. One might be sceptic about the uniformal blood Phe levels.
Question 2
Why was not increased the dose of saproterin from 10 mg to 20 mg in patients, who failed to achieve diet liberalization?
Suggestions:
The recently published revised European Guideline should be included in the introduction section, where the classification of PKU is detailed.
Author Response
REVIEWER 1
The manuscript deals with an important topic, it is of interest for PKU scientific community.
Comment 1. Table 1: How do you explain that the pre- and post-therapy blood Phe levels showed similar values in most patients (e.g. 484.29 or 726.44 micromol/L)? In reality, there are no such similar blood values in a PKU group. This is a significant question, because the main achievements were concluded from these blood results. One might be sceptic about the uniformal blood Phe levels.
Response 1. The similarity of some pre- and post-therapy Phe values reflects an analytical artifact rather than true biological uniformity. Some patients - particularly the oldest ones in our retrospective cohort - were at that time investigated with a semi-quantitative bacterial inhibition assay (Guthrie test), which reports integer results only (no decimals). In our laboratory these results were recorded in mg/dL. In order to harmonize these results to µmol/L, a standard conversion [mg/dL × 10,000/165.19 = µmol/L] was applied. Consequently, patients sharing the same integer mg/dL value, unrespectful of the decimal numbers, have identical µmol/L values (e.g., from 8.0 to 8.9 mg/dL → 484.29 µmol/L; from 12.0 to 12.9 mg/dL → 726.44 µmol/L). If the reviewer deems it appropriate, we could add this clarification to the Methods section.
Comment 2. Why was not increased the dose of saproterin from 10 mg to 20 mg in patients, who failed to achieve diet liberalization?
Response 2. The dose of Sapropterin was not increased beyond 10 mg because those patients (pt. 7 and pt. 13) were enrolled in an international clinical trial (KAMPER - Biomarin, trial number NCT01016392). According to the study protocol, Sapropterin could only be administered at doses of either 5 mg or 10 mg. Therefore, escalation to higher doses, such as 20 mg, was not permitted within the framework of the trial.
Comment 3. Suggestions: The recently published revised European Guideline should be included in the introduction section, where the classification of PKU is detailed.
Response 3. Thank you for this helpful suggestion. We have cited the revised European Guideline in the Introduction section (lines 75–80), adding the following text:
"According to the recently published first revision of the European Guidelines on PKU, classification should not rely solely on untreated Phe levels but also integrate responsiveness to cofactor therapy, reflecting a more nuanced and clinically relevant approach to patient management. Thus, patients should be classified as: (a) not requiring treatment (Phe <360 μmol/L); (b) requiring treatment, cofactor responsive; or (c) requiring treatment, cofactor unresponsive)."
GENERAL NOTE TO REVIEWERS
In addition, we have modified the title of the manuscript to: “Sapropterin Dihydrochloride Responsiveness in Phenylketonuria: A Case Series Exploring Gaps in Comprehensive Patient Monitoring”, since the term “Sapropterin” was missing in the previous version.
All modifications made to the manuscript have been highlighted in yellow for clarity.
Reviewer 2 Report
Comments and Suggestions for Authors
Thank you for submitting the manuscript "Dihydrochloride Responsiveness in Phenylketonuria: A Case Series Exploring Gaps in Comprehensive Patient Monitoring" to Nutrients.
Although the manuscript is a case record, the field of phenylketonuria is lacking in publications, as it is a small population, yet requires a significant exchange of information due to the difficulty of managing cases.
However, the manuscript reports retrospective experimental data evaluating a drug, which does not quite fit the scope of this journal and would be more appropriate in another MDPI journal. Therefore, in my opinion, to maintain the submission to Nutrients, the authors should consider including more results related to food or nutrition, and the focus of the manuscript should be on this. Furthermore, I have other points:
- It is very important to indicate who the authors need to revise the text, considering that sentences should not exceed two or three lines to avoid hindering comprehension. There is an excessive use of semicolons, which detracts from the clarity and flow of the text.
- Tables 1, 2, and 3 can be summarized using appropriate descriptive analysis, such as medians and quartiles. It is necessary to consider the best approach. The way these data were presented suggests that the manuscript was written in the form of a report. A more in-depth analysis of the text is needed to improve the quality of the technical writing.
- It is important to include, even as a supplementary file, the types of foods used in these patients' diets. Was a food frequency questionnaire previously administered? It would provide important information about the nutritional approach.
Author Response
REVIEWER 2
Thank you for submitting the manuscript "Dihydrochloride Responsiveness in Phenylketonuria: A Case Series Exploring Gaps in Comprehensive Patient Monitoring" to Nutrients.
Although the manuscript is a case record, the field of phenylketonuria is lacking in publications, as it is a small population, yet requires a significant exchange of information due to the difficulty of managing cases.
Comment 1. […] the manuscript reports retrospective experimental data evaluating a drug, which does not quite fit the scope of this journal and would be more appropriate in another MDPI journal. Therefore, in my opinion, to maintain the submission to Nutrients, the authors should consider including more results related to food or nutrition, and the focus of the manuscript should be on this.
Response 1. We appreciate the reviewer suggestion. In order to better align our manuscript with the scope of Nutrients, we have expanded the nutritional aspects of our study by explicitly describing the consequences on food intake and dietary management of patients before and after sapropterin treatment. Specifically:
- In the Methods section (lines 128–134; 146–147) we added: “Before sapropterin initiation, all patients were managed with an individually tailored Phe-restricted diet, in line with their metabolic tolerance. Dietary prescriptions included natural foods with low Phe content (mainly fruits and vegetables), with or without limited amounts of natural protein sources depending on individual tolerance. To ensure adequate protein and micronutrient intake, patients received Phe-free amino acid mixtures as protein substitutes, as well as multivitamin and long-chain polyunsaturated fatty acid (LC-PUFA) supplementation.” [...] “After initiation of Sapropterin, dietary management was adjusted according to the response to treatment.”
- In the Results section (lines 221–228) we specified: “Following treatment, 21 out of 23 patients achieved sufficient phenylalanine (Phe) tolerance to discontinue dietary restrictions. The remaining two patients exhibited partial responsiveness: patients 7 and 13 achieved tolerances of 1200 mg/day, (table 2) and continued on a partially Phe-restricted diet, consisting of about one-third Phe-free protein substitutes and two-thirds natural protein sources. In these two cases, the Sapropterin dosage was not increased beyond 10 mg/kg/day, as both were enrolled in the international KAMPER trial (BioMarin, NCT01016392), which restricted the maximum permitted dose to 10 mg/kg/day.”
- In the Discussion section (lines 400–405) we emphasized: “From a nutritional perspective, our findings underscore the substantial impact of sapropterin therapy on food intake and dietary management in PKU. Before treatment, all patients required a strict Phe-restricted diet, supported by Phe-free amino acid mixtures and micronutrient supplementation. Following responsiveness to Sapropterin, most patients were able to liberalize their diet, discontinue protein substitutes, and rely predominantly on natural food sources, thus improving dietary quality of life.”
We believe that these additions would directly address the reported concern by strengthening the nutritional perspective of our work and placing greater focus on the impact of Sapropterin therapy on dietary management in PKU.
Comment 2. It is very important to indicate who the authors need to revise the text, considering that sentences should not exceed two or three lines to avoid hindering comprehension. There is an excessive use of semicolons, which detracts from the clarity and flow of the text.
Response 2. Thank you for this helpful observation. We have revised the manuscript to improve readability by removing the excessive use of semicolons (e.g., lines 237 and 265) and by avoiding overly long sentences, which we separated into shorter ones for greater clarity (e.g., line 469).
Comment 3. Tables 1, 2, and 3 can be summarized using appropriate descriptive analysis, such as medians and quartiles. It is necessary to consider the best approach. The way these data were presented suggests that the manuscript was written in the form of a report. A more in-depth analysis of the text is needed to improve the quality of the technical writing.
Response 3. As kindly suggested, we have revised the descriptive analysis of our data using medians and interquartile ranges (IQR), which provide a more accurate summary given the non-homogeneous distribution of some variables, particularly pre-treatment phenylalanine levels, (see Table 1 and lines 202-203: “The median pre-test Phe level was 484.29 µmol/L (IQR 484.29-726.44), while at two weeks post-test it was 157.39 μmol/L (IQR 121.07-181.61).
Comment 4. It is important to include, even as a supplementary file, the types of foods used in these patients' diets. Was a food frequency questionnaire previously administered? It would provide important information about the nutritional approach.
Response 4. Thank you for this suggestion. As already suggested by Reviewer 1, we have integrated specific sections in the manuscript describing the dietary management of these patients (Methods, lines 128–134 and 146–147; Results, lines 221–228; Discussion, lines 400–405).
Regarding the second part of the comment, we clarified in the Methods section (lines 163–169) that:
"[…] all patients (or their parents, in the case of children) were accustomed to completing a daily food diary, which was systematically brought to each follow-up visit and reviewed together with the clinical team. These diaries included information on food choices, dietary preferences, and any issues related to the acceptance or palatability of Phe-free protein substitutes. This approach allowed clinicians to adapt the dietary regimen to the individual needs and tastes of each patient, ensuring both adherence and personalization of nutritional management."
This addition highlights that, although no pre-formulated questionnaire was used, detailed nutritional monitoring was systematically performed through food diaries.
GENERAL NOTE TO REVIEWERS
In addition, we have modified the title of the manuscript to: “Sapropterin Dihydrochloride Responsiveness in Phenylketonuria: A Case Series Exploring Gaps in Comprehensive Patient Monitoring”, since the term “Sapropterin” was missing in the previous version.
All modifications made to the manuscript have been highlighted in yellow for clarity.